# Effects of Oxidative Stress on Protein Translation: Implications for Cardiovascular Diseases

**DOI:** 10.3390/ijms21082661

**Published:** 2020-04-11

**Authors:** Arnab Ghosh, Natalia Shcherbik

**Affiliations:** Department for Cell Biology and Neuroscience, School of Osteopathic Medicine, Rowan University, 2 Medical Center Drive, Stratford, NJ 08084, USA

**Keywords:** protein translation, ribosome, RNA, IRES, uORF, miRNA, cardiovascular diseases, reactive oxygen species, oxidative stress, antioxidants

## Abstract

Cardiovascular diseases (CVDs) are a group of disorders that affect the heart and blood vessels. Due to their multifactorial nature and wide variation, CVDs are the leading cause of death worldwide. Understanding the molecular alterations leading to the development of heart and vessel pathologies is crucial for successfully treating and preventing CVDs. One of the causative factors of CVD etiology and progression is acute oxidative stress, a toxic condition characterized by elevated intracellular levels of reactive oxygen species (ROS). Left unabated, ROS can damage virtually any cellular component and affect essential biological processes, including protein synthesis. Defective or insufficient protein translation results in production of faulty protein products and disturbances of protein homeostasis, thus promoting pathologies. The relationships between translational dysregulation, ROS, and cardiovascular disorders will be examined in this review.

## 1. Introduction

The process of protein synthesis, or protein translation, constitutes the last and final step of the central dogma of molecular biology: assembly of polypeptides based on the information encoded by mRNAs. This complex process employs multiple essential players, including ribosomes, mRNAs, tRNAs, and numerous translational factors, enzymes, and regulatory proteins. To efficiently produce a faultless cellular proteome in a cell, protein translation is tightly controlled, aiming to establish protein homeostasis (proteostasis) to maintain healthy cellular physiology. Therefore, it is not surprising that any abnormalities in protein translation, caused by internal or external insults, that result in production of anomalous and potentially deleterious proteins, manifest in the development and progression of a plethora of human diseases. Many excellent reviews summarize the effects of translational errors and disturbed proteostasis on the development of neurodegenerative disorders [1,2], while cardiovascular diseases (CVDs), the leading cause of death worldwide (https://www.who.int/health-topics/cardiovascular-diseases), have received less attention. The goal of this review is to summarize our current knowledge of molecular alterations that occur within the translational machinery in response to reactive oxygen species (ROS) and lead to various cardiovascular pathologies.

## 2. ROS, Mitochondria and Electron Transport Chain, Oxidative Stress, and CVDs

### 2.1. ROS, ROS Sources and Sites

ROS is a common name for diverse chemical species, including superoxide anions, hydrogen peroxide, hypochlorous acid, nitric oxide, peroxynitrite, singlet oxygen, hydroperoxyl, and the hydroxyl radicals (Table 1), that are produced in cells as by-products of normal cellular metabolism. However, each of these molecules has its own unique characteristics, such as way of inactivation, substrate preferences, reactivity, kinetics, diffusion properties, etc. (reviewed in [3]).

ROS can target, terminally modify, and damage any biomolecule in a cell, such as lipids, proteins, and nucleic acids. At the same time, ROS may play a cytoprotective role by attacking invading pathogens or acting as signaling molecules. Signaling properties of ROS include activation of cellular kinases, inhibition of phosphatases, and upregulation of activities of certain transcriptional factors (reviewed in [4]). For example, transcription factors Yap1 from yeast and NRF2 from mammalian cells are capable of sensing low-dose H_2_O_2_. Upon detecting this ROS molecule, Yap1 and NRF2 translocate to the nucleus to promote synthesis of antioxidant defense enzymes [5,6], such as catalases, superoxide dismutases, thioredoxin-dependent peroxiredoxins, and glutathione peroxidases, all of which have their own substrate preferences and mechanisms of catalysis [7]. However, some of these defense enzymes exhibit pro-oxidant properties in pathological states. The best studied example is superoxide dismutase 1 (Sod1), also known as Cu/Zn dismutase. Under pathological conditions, such as atherosclerosis and hypertension [8] or upon mutation-mediated conformational rearrangements [9,10], Sod1 gains an ability to interact with H_2_O_2_ instead of O_2_^•–^ and promotes protein and lipid oxidation.

Among non-enzymatic antioxidants is the highly abundant, short (three amino acid) peptide glutathione (GSH), the intracellular concentration of which ranges between 1–10mM depending on the cell type [11]; vitamins C and E [12].

ROS can be supplied from the environment or from within the cell. Exogenous sources include exposure to radiation or high levels of transition metals, heavy metals, and several anticancer drugs [13]. The main intracellular origin of ROS is the mitochondria, but other cellular sites, such as the endoplasmic reticulum (ER), peroxisomes, and actin cytoskeleton, lysosome, plasma membrane, and cytosol are also known to supply ROS (reviewed in [14,15]). The main intracellular sources of ROS are the electron transport chain (ETC) of mitochondria (see below) and certain enzymes that generate ROS as reaction by-products. For example, xanthine oxidase (XO), which represents an oxidized form of xanthine dehydrogenase (XDH), forms in damaged tissues, including rat hearts with myocardial ischemia. XO can transfer electrons to molecular oxygen, resulting in the formation of superoxide during xanthine or hypoxanthine oxidation and promoting further heart damage [16]. Another example of enzymes that supply ROS is NADPH oxidases (NOXs), a unique family of enzymes the only function of which is producing ROS. The five family members (NOX 1–5) share the same structural characteristics, including a common catalytic core composed of two domains: the dehydrogenase domain and the transmembrane domain that ensures membrane localization of these enzymes. Electrons are transferred from cytosolic NADPH to FAD located in the dehydrogenase domain, then to the inner and outer heme present in the transmembrane domain, and finally to oxygen located on the opposite site of the membrane, thus generating either superoxide (NOX 1–3, NOX 5) or H_2_O_2_ (NOX 4) (reviewed in [17]). NOXs are implicated in CVD pathologies, including myocardial ischemia-reperfusion injuries, myocardial damage, and cardiomyocyte death [18,19].

### 2.2. Mitochondria and Electron Transport Chain

Mitochondria accomplish multiple functions critical for healthy cell physiology. Those include assembly of iron-sulfur clusters [20], respiration and energy production, regulation of Ca^2+^ content inside the cell, and fatty acid *β*-oxidation (reviewed in [14]). Of particular interest to this review is energy production via ETC. The ETC is composed of four multiprotein complexes (Complexes I-IV) embedded in the inner mitochondrial membrane (IMM), and two small electron carriers, ubiquinone (Q) and cytochrome (C), which shuttle electrons between the complexes. Complexes I and II transfer electrons to Q, from which electrons pass through Complex III, C, and Complex IV to oxygen. The Complex I/III/IV pathway uses NADH as the substrate, while Complex II/III/IV pathway utilizes succinic acid as the substrate. For detailed descriptions of ETC complexes, please read two recently published reviews [21,22]. Electron movement generates a proton gradient across the IMM, resulting in pumping protons (hydrogen ions, H^+^) from the mitochondrial matrix to the intermembrane space (IMS). This creates a force that drives protons back to the matrix through Complex V (ATP synthase), leading to ATP production [23], the process that is called oxidative phosphorylation [24].

However, not all electrons within the ETC are transferred to the final electron acceptor, O_2_, due to the phenomenon known as free-radical leak [25]. This results in production of mitochondrial ROS, mostly superoxide radical anion and H_2_O_2_, that were first detected several decades ago [26,27]. Further studies have unraveled ROS-generating sites within mitochondrial structures. Complexes I and III constitute two major origins of ROS, but succinate-mediated ROS production associated with Complex II has also been detected [28]. Thus, Complexes I, II, and III release O_2_^•–^ into mitochondrial matrix, while Complex III also releases O_2_^•–^ into IMS [29]. To avoid damages to mitochondrial and cytosolic macromolecules (in case ROS escape the mitochondria), cells have adapted two strategies. One is ROS scavenging by appropriate enzymes: for example, superoxide dismutases capture O_2_^•–^, while catalases and peroxiredoxins inactivate H_2_O_2_ [7]. The second possible strategy is minimizing free radicals leak by increasing expression of respiratory complexes. However, this process requires fine-tuning of mitochondrial and nuclear genes expression, as components of ETC are encoded by both the nuclear and mitochondrial genomes, which are prone to mismatch [25].

### 2.3. Oxidative Stress and CVDs

Nevertheless, in situations when defense systems are either overwhelmed or malfunction, upon abnormalities/mutations in ETC components, or during loss of mitochondrial inner membrane integrity, ROS levels exceed functional capacity of antioxidants [15], creating a condition called oxidative stress. In fact, the concept of oxidative stress was first formulated in 1985 as “a disturbance in the prooxidant-antioxidant balance in favor of the former” [30]. Three decades of extensive research have built a body of evidence establishing oxidative stress as a complex and multifaceted physiological condition that can be divided into several sub forms, ranging from physiological to toxic and acute (for more details, please read an excellent review written by Helmut Sies and colleagues [31]).

In relation to this review, numerous studies have provided evidence that acute oxidative stress is one of the major causes of CVDs (reviewed in [32]). As such, excessive amounts of ROS contribute to heart failure [33], atrial fibrillation [34], hypertension [35,36], and atherosclerosis [37]. Therefore, it is not surprising that various ROS-scavenging strategies have been developed to treat CVDs. However, certain challenges still exist (discussed in Section 7 of this review). The cross-connections between oxidative stress and other type of stresses, such as hypoxia [38], ER stress [39,40], and nutritional stress or obesity are also noteworthy [41]. Thus, any abnormalities in the functions of such a complex biosystem as the human body may ultimately reflect on the well-being of the cardiovascular system. Understanding the origin of the problem on the molecular level may help in developing new drugs specifically targeting malfunctioning molecular segments, treating and even preventing CVD development.

## 3. Overview of the Translational Steps in Eukaryotes

Eukaryotic translation proceeds through intricate but well-defined stages of initiation, elongation, termination, and ribosome recycling, illustrated in detail in Figure 1. At the end of a single round of translation, ribosomal subunits dissociate and are recycled to participate in the next round (Figure 1, top left). Eukaryotic initiation factors, eIF1 and eIF1A, and the ATPase binding cassette protein ABCE1 are involved in the recycling process [42]. eIF6 binds to the large 60S subunit of the ribosome to prevent its premature association with the small 40S ribosomal subunit [43]. Similarly, a large eIF3 complex (composed of 13 subunits in mammals, ~790 kDa) bound to the interface side of the small subunit also impedes its association with the 60S [44]. Aminoacylated tRNA_i_ (Met-tRNA_i_^Met^) binds to the small subunit as part of the ternary complex (TC) containing eukaryotic initiation factor 2 (eIF2) bound to GTP (TC: eIF2 bound to GTP and Met-tRNA_i_^Met^). The resulting 40S•eIF1•eIF1A•eIF3•TC (43S) pre-initiation complex (Figure 1, bottom left) attaches to the capped 5ʹ end of the message primarily through interactions between the scaffolding protein eIF4G and eIF3 resulting in the 48S complex (Figure 1, bottom). eIF5 also plays a role in recruiting 43S to the mRNA [45].

eIF4G remains associated with the 5′ end cap-binding protein eIF4E and the ATP-dependent helicase eIF4A. The 48S complex thus forms and scans the structured 5′ untranslated region (5′UTR) of the mRNA in search of the correct start codon. The 3ʹ end of the message bound to the poly(A)-binding protein interacts with the cap-binding proteins and maintains communication between the 5′ and 3′ ends of the mRNA [46]. During scanning, the positioning of an AUG start codon in the P-site of the 40S results in structural rearrangements within the ribosome. These structural reorganizations lead to stable binding of Met-tRNA_i_^Met^ to the AUG start codon of the mRNA (Figure 1, bottom right) [45,47]. The majority of the eIFs dissociates from the ribosomal complex except for eIF1A. The large ribosomal subunit bound to eIF5B•GTP now attaches to the small subunit followed by hydrolysis of eIF5B•GTP, while eIF5B•GDP and eIF1A dissociate, causing the formation of elongation-competent 80S complex (Figure 1, right) [48].

During the elongation phase, aminoacylated tRNAs are delivered to the A-site of the ribosome by the eukaryotic elongation factor 1A (eEF1A) in a GTP-dependent manner. Correct codon-anticodon base pairing results in the eEF1A-bound GTP hydrolysis ensuing dissociation of the factor from the ribosome and tRNA accommodation into the A-site (Figure 1, top right). eEF1B is another important factor involved in recycling the eEF1A•GDP with eEF1A•GTP and preparing it for the next round of delivering aminoacylated tRNA into the A-site [49]. The 80S ribosome decodes the message and forms peptide bonds between two amino acids at the peptidyl transferase center of 80S. The ribosome translocates one codon towards the 3′ end of the mRNA so that the A-site becomes available to accommodate a new aminoacylated tRNA. Each step of the translation cycle adds a new amino acid to the growing nascent peptide chain, which moves through the exit tunnel that transverses the large subunit 60S. The elongation phase continues until the elongating ribosome encounters a stop codon at the A-site, signaling termination of translation (Figure 1, top). Translation termination involves two different classes of release factors. Eukaryotic release factor 1 (eRF1) (class I) recognizes the stop codons and is responsible for hydrolysis and release of the nascent peptide from the ribosomal complex. Class II factor eRF3 is a ribosome-dependent GTPase that stimulates peptide release and stop codon recognition efficiency in a GTP-dependent manner [50].

Oxidative stress-mediated damage or modifications to ribosomal subunits, mRNA, tRNA, and the various factors and enzymes involved in translation regulation could affect multiple translation steps (Figure 1, red stars), leading to altered levels of gene expression. This, in turn, might have detrimental consequences on the proper functioning of the cardiovascular system.

## 4. Translation Initiation in Pathophysiology of CVDs

Of the four steps of translation, the majority of regulation occurs during the initiation step, affecting translation efficiency or capacity. In this section, we will focus on the current knowledge of regulation of translation initiation via canonical and non-canonical mechanisms. Implications in etiology and development of CVDs and connection to oxidative stress will be also discussed.

### 4.1. Translation Initiation Factor eIF2

The most well-studied effect of oxidative stress-induced regulation of translation initiation is via eIF2, a heterotrimeric complex composed of 3 subunits: α, β, and γ. Phosphorylation of the α subunit at the Ser-51 residue, which occurs in response to stress, reduces the ability of eIF2 to bind GTP and deliver the initiator Met-tRNA_i_^Met^ to the small ribosomal subunit (Figure 1, left). Thus, eIF2 transmits a stress signal to the translational machinery via phosphorylation of eIF2α and represents the primary regulatory switch of the canonical translation initiation mechanism, resulting in global repression of translation [51]. In mammalian cells, eIF2α is phosphorylated by four different kinases, which are activated by different types of stresses. PKR kinase is activated by the presence of double-stranded RNA; amino acid starvation induces GCN2 kinase; ER stress turns on PERK kinase activity; and HRI kinase is activated by heme deficiency, heat shock, oxidative stress, and osmotic shock [52]. Additionally, in yeast, ROS result in the phosphorylation of eIF2α via activation of the yeast amino acid control kinase Gcn2 [51].

Recent studies investigating the potential roles of these kinases have yielded some interesting insights into possible connections between cellular stress in cardiomyocytes and cardiac anomalies. Studies using *GCN2^−^*^/*−*^ mice have shown that the *GCN2* deletion itself did not result in any noticeable changes in cardiac structure or primary cardiac function. However, when subjected to chronic cardiac pressure overload through transverse aortic constriction, cardiac failure was diminished in *GCN2^−^*^/*−*^ mice when compared with the control mice [53]. A substantial reduction in ventricular dysfunction and ventricular dilation, associated with pulmonary congestion, was observed. Interestingly, loss of GCN2 function resulted in increased expression of the anti-apoptotic factor Bcl-2, along with a reduction in both oxidative stress and cardiomyocyte cell death via apoptosis [53]. Another study conducted on diabetic *GCN2^−^*^/*−*^ mice demonstrated reduction in lipotoxicity and oxidative stress [54]. Together, these studies provide an explanation for the cardioprotective role of *GCN2* deletion. On the contrary, in vivo treatment of male C57Bl/6J mice and human cardiac fibroblasts with halofuginone, a bioactive compound that induces the GCN2-mediated stress response pathway, also resulted in improved survival, ventricular function, and reduced ischemic injury in mice [55]. These contradictory findings most likely stem from the fact that different experimental systems were used [53,54,55]. It is also possible that, in addition to activating the GCN2-stress response pathway, halofuginone may have off-target effects; thus, these data cannot be directly compared to the effects of a permanent deletion of GCN2 on the cardiac system.

To understand the role of eIF2α-PERK in processes leading to heart failure, Liu and co-authors subjected wild-type and PERK*^−^*^/*−*^ mice to transverse aortic constriction [56]. Knock-down mice demonstrated abrogation of ventricular remodeling and were more susceptible to cardiac failure than control mice, suggesting that PERK plays a critical role in cardiomyocyte adaptation to chronic pressure overload-induced stress, thereby protecting the heart [56]. Opposite results were obtained with another kinase of eIF2α: PKR knockout mice showed a reduction in pulmonary congestion and myocardial fibrosis compared to the wild-type controls upon cardiac pressure overload [57]. In addition, increased eIF2α phosphorylation was detected in human patients experiencing heart failure, and in the Friedreich ataxia mouse model, characterized by reduced cardiac function [58,59]. Finally, deacetylation of eIF2α by SIRT1 was reported to protect cardiomyocytes from ER stress-induced injuries [60]. Taken together, these studies demonstrate that the translation factor eIF2 plays a differential role in cardiac survival and adaptation to various stress conditions, a phenomenon that is likely controlled by different stress-induced kinases. Clearly, further studies are required to better understand the molecular mechanisms that regulate stress-dependent phosphorylation of eIF2α in cardiac tissues and physiological connections between the efficiency of translation initiation and cardiac anomalies.

How do ROS impact eIF2 activity on the molecular level? As discussed above, it is the commonly accepted view that, under various stress conditions, the α subunit of eIF2 undergoes phosphorylation leading to a global downregulation of conventional cap-dependent translation initiation. Within this dogma, the best-studied and most straightforward molecular mechanism is based on the activation of ER-related mammalian PERK and yeast Ire1 kinases [61]. Thus, massive misfolding of ER proteins enforced by ER stressors, including ROS, leads to ER stress, PERK/Ire1 activation, and eIF2α phosphorylation. It is worth mentioning that ER stressors alone cause ROS formation, exemplified by detection of H_2_O_2_ in tunicamycin-treated cells [62]. This mechanism demonstrates an indirect role of ROS in inhibiting eIF2 activity via phosphorylation. However, one of the unexplored possibilities is that eIF2 might be a direct target of ROS. For example, reversible oxidation of thiols present on cysteine residues of eIF2 subunits might establish an additional regulatory mechanism that utilizes Cys switches [63]. In fact, using genome-wide quantitative redoxome analysis OxiCAT [64], Topf and colleagues (2018) identified eIF2α/SUI2 and eIF2β/SUI3 as potential targets of thiol oxidation by H_2_O_2_ in yeast. Thus, the H_2_O_2_-sensitive Zn^2+^-coordinating sequence (CX_2_C-X_19_-CX_2_C) was found within eIF2β/SUI3 [65], suggesting that the β subunit of eIF2 might represent the target of H_2_O_2_-dependent regulation via Cys-switch. Similarly to β, the α subunit of eIF2 was also among the proteins containing putative H_2_O_2_-sensitive cysteines (Table 5 in [65]). Interestingly, the same study demonstrated that phosphorylation of eIF2α from both yeast and mammalian HEK-293 cells was decreased upon acute H_2_O_2_-mediated oxidative stress, while translation remained suppressed. This observation demonstrates that attenuating eIF2-dependent translation initiation utilizes multiple mechanisms, of which phosphorylation-dependent inhibition of eIF2 constitutes only one. This raises the possibility that an alternative way to inhibit eIF2 activity might be via direct oxidation of eIF2 subunits by ROS, in particular H_2_O_2_. Whether these new findings contribute to cardiopathologies remains to be determined.

### 4.2. Non-Canonical Regulation of Translation Initiation and Cardiovascular Pathologies

In addition to the canonical model of translation initiation (summarized in Section 3), ribosomes can also initiate translation in non-conventional ways utilizing internal ribosome entry sites (IRES) or upstream open reading frames (uORFs), both of which are located in the 5ʹUTR (Figure 2A,B).

#### 4.2.1. IRESs

IRES are highly structured elements that directly recruit ribosomes and initiate translation from a start codon in a scanning-independent manner (Figure 2A). IRES possess minimal or no requirements for canonical eIFs, especially the eIF4 cap-binding proteins. Alternate initiation utilizing these elements allows cells to regulate gene expression under different conditions [66,67]. First discovered in vertebrate-infecting RNA viruses like picornaviruses and hepatitis C virus (HCV), IRESs govern preferential expression of viral proteins.

The *Cardiovirus* genus within the *Picornaviridae* family comprises positive-sense, single-stranded RNA (ssRNA) viruses known to attack cells of the host’s cardiovascular system and cause myocarditis, leading to arrhythmias and heart failure [68,69]. Picornavirus RNA lacks the 5ʹ cap, and its expression is driven by an IRES element located in the 5ʹUTR of the viral mRNA [70]. Despite the sequence variability of the 5ʹUTR, most of the RNA structural motifs are largely conserved in different picornaviruses [71]. The 2A protein of encephalomyocarditis virus contains a nuclear localization signal-like sequence in its carboxy-terminal end, which allows import into the host’s nucleus and interaction with the ribosome. Stable binding of 2A with 40S prevents cap-dependent translation initiation of cellular RNAs, thereby favoring IRES-mediated expression of the viral RNA [72]. Other cardioviruses that infect humans include Saffold virus, Syr-Darya Valley fever virus, and Vilyuisk human encephalitis virus [73,74,75,76], but the presence of IRES elements within the ssRNA of these viruses remains unknown.

Chronic infection by HCV (family *Flaviviridae*) is also associated with CVDs [77,78]. Similar to cardioviruses, HCV RNA expression is driven by IRES-mediated translation initiation [77]. Under cellular stress conditions, specific domains of HCV IRES interact with the small ribosomal subunit and the initiation factors eIF3 and eIF5B (Figure 1, left), allowing expression of viral RNA [79]. In addition, some viruses from other families unrelated to cardioviruses are known to target cells of the cardiovascular system and contribute to the etiology and progression of CVDs (for a detailed review, see [80]). These examples demonstrate that cardiovascular pathologies may develop in response to viral infections, and that viral propagation inside the cell is often driven by unconventional IRES-dependent translation initiation mechanisms.

Interestingly, IRES-mediated translation initiation is not limited only to viruses. Several cellular mRNAs were found to contain IRESs in their 5ʹUTR in addition to the 5ʹ cap, implying that their translation can initiate through both cap-dependent and cap-independent mechanisms. In addition, different IRES elements interact with other trans-acting factors during initiation [81,82].

The best-studied example of IRES-mediated translation initiation of cellular genes includes the angiogenic vascular endothelial growth factor A (VEGF-A). VEGF-A plays a crucial role in processes of new blood vessel growth: angiogenesis, arteriogenesis, and vasculogenesis. Loss of a single allele or low expression of this factor results in early embryonic lethality [83,84], while defective or dysfunctional vascularization is a significant component of pathophysiology of heart failure in adults, which is associated with hypertension, ischemic heart disease, and valvular abnormalities (reviewed in details in [85]). Constantly expressed at basal levels, VEGF-A protects endothelial cells from apoptosis, but increased expression of this factor has been detected in various cardiopathologies [86]. This suggests that tight regulatory control of VEGF-A expression is critical, at least in cardiac cells. VEGF-A is generated in five major isoforms, all of which are secreted from different kind of cells, including cardiomyocytes [87]. VEGF-A expression is controlled on several levels, including transcriptional activation by Sp1/Sp3 factors, which, under oxidative stress, interact with the VEGF-A promoter [88]. Interestingly, two independent IRES elements (IRES-A and IRES-B) were identified at the 5ʹUTR region of *VEGF* [89,90,91] and linked to the regulation of this factor’s expression in response to hypoxia [92], which promotes ROS formation and oxidative stress [37].

How do ROS control VEGF at the molecular level? The cornerstone in the ROS-VEGF pathway is H_2_O_2_, derived from superoxide during dismutation. Superoxide can be supplied by NOX or released from the mitochondria. In turn, H_2_O_2_ can be either neutralized in a two-step reaction catalyzed by catalase [93], or generate oxidizing radicals able to oxidize lipids and proteins, including hypochlorous acid (HOCl, generated by myeloperoxidase using H_2_O_2_ as a substrate [94]), or OH• (via the Fenton reaction [95]). To investigate the involvement of VEGF in CVD pathophysiology, Ruef and co-authors treated smooth muscle cells with H_2_O_2_ or with its derivative 4-HNE (a lipid peroxidation product present in atherosclerotic lesions) and examined levels and localization of VEGF. Both treatments demonstrated upregulation of VEGF protein that was efficiently secreted from the cells [96]. Similar results were obtained with rat heart-derived endothelial cells, in which H_2_O_2_ was found to be a strong inducer of VEGF production and secretion [97]. Both studies provided evidence that H_2_O_2_-dependent upregulation of VEGF expression occurs at the transcriptional level, suggesting that ROS, in particular H_2_O_2_, likely contributes to the upregulation of the *VEGF* promoter, as has been discussed earlier for *VEGF-A* and Sp1/Sp3 [88]. Given that hypoxia was found to strongly induce VEGF expression both *in vitro* and *in vivo* [98], it is possible that hypoxia-inducible factors (HIFs), which respond to decreased oxygen availability, might control *VEGF* transcription in cardiac cells in response to elevated levels of ROS in hypoxia, in addition to IRES-mediated translation initiation.

Another example of IRES-mediated translation initiation that occurs in cardiomyocytes in response to stress is expression of the cytokine fibroblast growth factor 1 (FGF-1), which belongs to a family of growth factors linked to CVDs [99]. FGF-1 mRNA contains four different leader sequences (A, B, C, and D) that are strikingly conserved among mammals; IRES elements have been identified in 5ʹUTRs of A and C [100]. Further studies implicated IRES trans-acting factor Vasohibin 1 in regulating IRES-mediated expression of FGF-1 in cardiomyocytes in response to hypoxia [101]. Similar hypoxia-induced, IRES-dependent translation was documented for Staufen-1 (Stau-1) [102] and hypoxia-inducible factor 1 (HIF-1) [103]. Stau-1 is the double-stranded RNA-binding protein involved in localizing mRNAs to proper cellular compartments (discussed in [104]). It is implicated in myotonic dystrophy type I, a multisystem disorder that, among other dysregulation, affects the cardiovascular system. The transcription factor HIF-1 is a vital regulator of oxygen homeostasis in metazoans, also implicated in pathophysiology of numerous cardiovascular diseases [105].

The ability of IRES elements to initiate translation with minimal requirements for canonical factors has prompted researchers to target different cardiovascular pathologies through gene therapy [106,107].

#### 4.2.2. uORF

uORFs are another example of non-canonical regulation of translation initiation that controls gene expression. Similar to IRES, uORFs are located at the 5ʹUTR of an mRNA, but unlike IRESs, uORFs require members of the canonical translation-initiation machinery (discussed in Section 3). However, by initiating translation, uORFs lead to reduced expression of the downstream sequence of the primary gene-coding ORF (Figure 2B, reviewed in [108]). More than 50% of transcripts in humans contain uORFs, thereby providing a mechanism of translational control [109] that seems to be an important component of the eukaryotic stress response [108]. One typical example of such regulation is the expression of ATF4 in mammals. This basic leucine-zipper transcriptional regulator contains two uORFs; the first uORF (uORF1) encodes a 3-amino acid peptide, while the second uORF (uORF2, 177 nucleotides) overlaps with the first 83 nucleotides of the *ATF4*-coding sequence (Figure 2C). Under normal, non-stressed conditions, translation initiated at uORF1 generates a short peptide, while 40S ribosomal subunits continue scanning mRNA and reassemble with eIF2•GTP-Met-tRNA_i_^Met^ at the out-of-frame uORF2’s initiation codon leading to translation termination (Figure 2C-1). Thus, uORF2 plays an inhibitory role by blocking ATF4 expression. During stress, eIF2 undergoes phosphorylation and requires extra time to reassemble the eIF2•GTP-Met-tRNA_i_^Met^ complex. This delay allows the mRNA-scanning 40S subunits to pass through the inhibitory uORF2, leading to translation initiation at the *ATF4*-coding ATG (Figure 2C-2). This elegant mechanism, called “delayed translation initiation”, results in the production of ATF4, which regulates transcription of numerous genes involved in signaling pathways and participates in response to various stressors [110]. However, the biological role of ATF4 during stress response depends on variety of factors and ranges from cytoprotective to pro-apoptotic [111]. Although ATF4 is viewed as stress-defense facilitator [112], several studies have reported the contribution of ATF4 to cardiovascular pathologies during oxidative stress. As such, in the rat model of transverse aortic constriction, the pathological stimuli-induced overexpression of ATF4 and increased ROS levels were found to contribute to cardiac fibroblast activation, the cellular reprogramming process that causes cardiac hypertrophy [113]. To detect ROS, the authors used 2ʹ,7ʹ-dichlorofluorescein diacetate (H2DCFDA), the fluorogenic dye that measures a wide range of ROS, including hydroxyl, peroxyl, and other species within the cell [114]; thus, the particular ROS specie(s) modulating ATF4′s activity is unknown. In another study, ATF4 expression in response to oxidative stress in human arterial cardiomyocytes leads to cardiomyocytes apoptosis [115] and is considered one of the causative factors of atrial remodeling and disease progression [116]. Although large progress has been made towards understanding ATF4 function in response to oxidative stress (reviewed in [117]), many questions remain unanswered. Thus, it is unclear whether a particular ROS specie(s) directly modifies ATF4 or acts indirectly by, for example, promoting uORF-dependent dysregulation of *ATF4* expression. Interestingly, phosphorylation of eIF2α by various kinases at Ser51 (see Section 4.1) leads to selective translation of ATF4 while global translation is inhibited [118], suggesting an important role for ATF4 during stress response. Understanding how ROS affect ATF4 will help in deciphering the molecular mechanisms of this factor’s activity in human pathologies, including CVD.

Titin (TTN) is the largest human protein that resides within the heart muscle and plays a critical role in maintaining the contractile activity of cardiac sarcomere. *TTN* undergoes alternative splicing, generating two major isoforms: N2B and N2BA. Mutations in this gene cause cardiac abnormalities, including dilated cardiomyopathy, which manifests as systolic dysfunction and dilation of the left ventricle. Several excellent reviews describe genetic variants of *TTN* and their effects on the etiology and progression of cardiomyopathies [119,120]. In addition, TNN was recently (2020) found to represent a heart-specific protein from the “cysteine redox network” in the Oximouse model [121]. It was demonstrated that TNN is prone to significant modifications via thiol oxidation during aging, leading to cardiomyopathies (for details see [121]). To understand whether TTN expression can be controlled by uORFs, Cadar and co-authors investigated the 5ʹUTR of *TTN* and identified two uORFs that suppressed TTN translation in non-stressed cardiac HL-1 cell line and primary neonatal rat ventricular myocytes [122]. Interestingly, *TTN* uORFs responded differently to distinct oxidative stressors: doxorubicin [123], but not H_2_O_2_, increased *TTN*-uORF-mediated translation efficiency [122], suggesting the possibility of various response mechanisms of cardiac cells to ROS.

As discussed in Section 4.2.1, the 5ʹUTR of the angiogenic factor VEGF-A contains two IRES elements. Interestingly, IRES-A was found to possesses a unique short uORF that inhibits expression of the VEGF121 isoform but not VEGF165 or VEGF189 isoforms of the factor, suggesting its role as a *cis*-element for isoform expression [124]. Whether and how ROS influence uORF activity of this gene expression remains to be found.

Dysfunctions in TGFβ (transforming growth factor β) cytokine activities contribute to arrhythmogenic right ventricular cardiomyopathy, a progressive myocardial disease that manifests as sudden death among young adult population [125]. TGFβ contains 11 predicted uORFs at the 5ʹUTR [126]. Two independent groups revealed that mutations in uORF-AUGs promote activation of TGFβ translation [126,127]. TGFβ is generated as large inactive complex composed of mature dimeric TGFβ molecules bound to latency-associated protein (LAP) and latent TGFβ-binding protein (LTBP). Activation of TGFβ involves release from LAP, process that is controlled by multiple mechanisms, including oxidative modifications of LAP or matrix metalloproteases (MMP) that cleave LAP. Both mechanisms activate TGFβ by releasing it from the latent complex (reviewed in [128]). Multiple studies have implicated various ROS in TGFβ activation process, including asbestos-derived ROS [129], nitric oxide [130], and intracellular or extracellular superoxide anion generated by Nox4 or Nox2, respectively [128]. There is interesting interplay between TGFβ and ROS: ROS activate TGFβ, while TGFβ enhances ROS levels by promoting their production and inhibiting antioxidants-defensive systems (reviewed in [131]). It is, therefore, tempting to propose an existence of a regulatory feed-back loop that would maintain physiological levels of TGFβ in cardiac cells via uORF-regulated translation in response to ROS levels.

uORF-mediated translation regulation has been identified for several other proteins and factors involved in CVDs etiology and progression. For example, differential expression of various Connexin 43 isoforms, varying only in their 5ʹUTR lengths, affects the formation of gap junctions in aged heart tissue [132]. The uORF of estrogen receptor α regulates its expression in a tissue-specific manner [133]. Similarly, the translational control of argininosuccinate synthase via its uORF affects nitric oxide levels and severity of oxidative stress [134].

### 4.3. Regulation of Translation Initiation by the mTOR Signaling Pathway

Oxidative stress targets different cellular signaling pathways, including the mammalian target of rapamycin (mTOR), which plays a crucial role in stress response and acts via two multiprotein complexes: mTORC1 and mTORC2 [135]. mTORC1 modulates the efficiency of translation initiation by targeting S6K1, eIF4B, and eIF4E-binding protein 1 (4E-BP1) as part of the complex cell signaling pathway. Phosphorylation of S6K1 by mTORC1 promotes translation initiation by (i) activation of cap-binding factor eIF4B, and (ii) inhibition of negative translation initiation regulators, like the programmed cell death 4 (PDCD4) protein [135]. In its unmodified form, 4E-BP1 binds to eIF4E, blocks its interaction with eIF4G, and downregulates the canonical mechanism of translation initiation. However, mTORC1-mediated phosphorylation of 4E-BP1 prevents eIF4E binding, resulting in eIF4G recruitment to the 5ʹ capped end of the message and efficient translation initiation [136].

Although mTOR is well known to play an essential role in cell growth regulation, whether mTOR activation has a positive or negative effect on the cardiovascular system remains debatable. Tamai and co-authors attempted to dissect the role of the mTOR pathway in early postnatal cardiac development using mice lacking Rheb [137], one of the critical upstream regulators of mTORC1. Heart-specific deletion of Rheb (Rheb^−/−^) caused lethality between days 8–10 of the postnatal period, with mice exhibiting impaired sarcomere maturation, reduced mRNA translation, and decreased S6 and 4E-BP1 phosphorylation [137]. This suggested that Rheb-mediated mTORC1 activation is required for proper cardiomyocyte development in the postnatal period. Consistent with this study, cardiomyocyte-specific deletion of mTOR of mTORC1 in the murine model led to compromised heart development and embryonic lethality [138]. Conversely, in another study, mutant mice with two hypomorphic (mTOR^Δ/Δ^) alleles expressed ~25% of wild-type mTOR levels and were not only viable, but had longer lifespans [139]. However, a decline in bone volume and immune function were also noticed, suggesting a tissue-specific effect of mTOR [139]. One possibility that would explain these contradictory findings is that the members of the mTOR signaling pathway undergo dynamic interplay that affects multiple facets of the cardiovascular system differently. In support of this possibility, moderate inactivation of mTORC1 by mTOR-inhibitor rapamycin was found to be cardioprotective, while genetic deletion of mTORC1 in mouse hearts produced deficiencies in cardiac development and function [140].

## 5. Impact of Oxidative Stress on RNA and RNA-Containing Complexes and Cardiopathologies

### 5.1. Ribosome

The eukaryotic ribosome is a massive (3.2 MDa) ribonucleoprotein complex composed of two subunits: small 40S and large 60S. Ribosomal subunits are built of ribosomal RNAs (rRNAs in mammals are 18S, 28S, 5.8S, and 5S) and numerous ribosomal proteins (r-proteins). rRNAs constitute a structural and functional core of the ribosome, while r-proteins fill gaps and cracks in the rRNA scaffold, making the ribosome a very compact yet extremely flexible, stable, and long-lived complex. Due to their high abundance and RNA-protein composition, ribosomes are susceptible to ROS-induced modifications, some of which severely affect ribosome integrity and function (reviewed in [141]).

Although oxidative stress is widely recognized as a significant driving force of CVD [142], the connection between CVD and oxidized ribosomes remains mostly unexplored. Searching for direct evidence of ribosome malfunction in CVDs upon RNA oxidation, Martinet and co-authors performed biochemical characterization of RNA extracted from carotid endarterectomy specimens derived from atherosclerotic plaques of 20 patients and compared them to 20 control segments of non-atherosclerotic mammary arteries [143]. The analysis revealed extensive fragmentation of mature 18S and 28S rRNAs in cells derived from plaques, while ribosomes purified from control arteries contained intact rRNA species. Furthermore, RNA extracted from atherosclerotic plaques was heavily modified with the oxidative stress marker 8-oxoguanosine (8-oxo-G), suggesting the possibility that oxidized ribosomes undergo degradation and, thus, negatively affect smooth muscle and endothelial cell survival and proliferation, leading to atherosclerotic plaque destabilization and rupture [144].

Several r-proteins were identified as causative factors of various CVDs, including r-protein Rps15, implicated in the progression of cardiomyopathy [145], and Rpl17, malfunctions of which have been linked to the development of carotid atherosclerotic vascular disease [146]. What makes these r-proteins pro-pathogenic remains unanswered. Considering that cardiovascular pathologies are associated with high levels of ROS and r-proteins are prone to ROS-driven modifications [141], it is tempting to speculate that ribosomal changes associated with oxidation-dependent modifications of r-proteins might play a role in cardiomyopathy and atherosclerosis. Thus, it will be interesting to examine and compare post-translational modification present on Rps15 and Rpl17 in healthy and diseased cardiac tissues. Consistent with previous studies [145,146], Rpl13A promoted the progression of hyperlipidemia, a condition associated with atherosclerosis, heart failure, and underlying metabolic disorders. In this case, Rpl13A was found to possess pro-oxidative stress characteristics, as disrupting one *rpl13a* allele in Chinese hamster ovarian cells conferred cell resistance to lipid-induced oxidative stress [147]. It is, however, unclear whether such an atypical capacity of r-protein derives from its activity within a ribosome or reflects a new function as an extra-ribosomal protein [148]. Although direct evidence is missing, these examples demonstrate an intricate connection between abnormal function of r-proteins in different cardiovascular pathologies that are associated with high levels of ROS.

Cardiovascular diseases are highly connected to metabolic disorders, including diabetes mellitus (DM). In fact, DM is proven to be associated with increased coronary artery, cerebrovascular, and peripheral vascular disease, causing a high mortality rate [149]. One of the prevalent molecular theories in the etiology of DM is based on the accumulation of ROS, as overproduction of ROS appeared to be a common pathogenic factor in insulin resistance and β-cells dysfunction. Studies conducted in the 1980s documented crosstalk between ribosomes, diabetes, and heart pathology. As such, a high rate of ribosome degradation in muscular and heart tissues was detected in diabetic rats post-insulin withdrawal [150,151]. Whether DM-associated ribosome decay is associated with ROS-induced damages to ribosomes awaits further investigations.

### 5.2. mRNAs and tRNAs

Unlike rRNAs, mRNAs and tRNAs are not protected by interaction with proteins and, therefore, are more susceptible to oxidative damage. The nitrogen and oxygen atoms in mRNA and tRNA nucleobases are mostly exposed triggering oxidative modifications, as opposed to double-stranded DNA, where they are relatively protected [152]. The highly reactive OH• radicals, generated during the oxidative stress reactions cascade, can interact with a wide variety of substrates, including free nucleobases, thus causing numerous modifications to RNA. The 8-oxo-G is the most notable RNA modification due to its drastic effect on RNA structure. Tanaka and co-authors showed that 8-oxo-G RNA modifications promote the release of cytochrome c from mitochondria, inducing apoptosis [153]. Another study detected a correlation between increased levels of 8-oxo-G modified RNAs in heart tissues of male Dahl salt-sensitive rats and development of heart failure symptoms, including increased blood pressure and myocardial hypertrophy [154]. Shan and co-authors demonstrated that oxidation of mRNA reduces protein synthesis and is an early marker for cell death in neurons of Alzheimer’s disease patients [155], suggesting that similar pathology may exist in terminally differentiated cardiomyocytes.

Oxidative modifications of mRNA can also impair base pairing within RNA and are capable of triggering ribosome quality control pathways [156]. Although oxidized mRNAs can recruit the translation apparatus for decoding, they eventually cause synthesis of erroneous protein products and premature translation termination in rabbit reticulocyte lysate and human HEK-293 cells [157]. Simms and co-authors showed that the 8-oxo-G modification on mRNA stalls the translation machinery, resulting in >3-fold reduction of protein synthesis. Such stalled products are substrates of the No-Go Decay (NGD) RNA surveillance pathway [158]. Although NGD removes damaged mRNAs efficiently, it can be overwhelmed by high production of damaged mRNAs.

RNA molecules with longer half-lives, like tRNAs, are even more susceptible to oxidative damage than other RNAs [159]. Oxidative modification of tRNA can lead to severe functional consequences, including altered base pairing with mRNA during translation, compromised stability, and miscoding [160]. Additionally, heightened oxidative stress can increase mutations in tRNA genes. Such mutations of mitochondrial tRNA genes are related to numerous cardiomyopathy-associated pathological conditions like mitochondrial encephalomyopathy with lactic acidosis and stroke-like episodes, myoclonic epilepsy with ragged red fibers, and maternally inherited diabetes and deafness [161]. Similarly, a mitochondrial mutation resulting in a m^1^G37 modification of tRNA^Met^ is related to maternally inherited hypertension [162]. A genetic study of nine infants belonging to five different families and suffering from severe cardiomyopathy revealed mutations in the GatCAB aminoacyl-tRNA amidotransferase complex, which charges the mt-tRNA^Gln^ via a transamidation reaction [163]. Instability of mitochondrial alanyl-tRNA synthetase (mt-AlaRS) causes fatal infantile cardiomyopathy [164]. Restoring tRNA^Ala^ levels in the affected heart using mitochondrial-targeted transcription activator-like effector nucleases reverts specific disease-related phenotypes in mice, opening up exciting therapeutic avenues against cardiovascular diseases [165]. Apart from these examples, evidence suggests that stress-induced tRNA modifications could also lead to altered gene expression profiles in eukaryotes. For example, cells can modify tRNA wobble bases and modulate gene expression of some codon-biased messages in response to stress [166,167]. Such modulation of gene expression profiles could have crucial implications on the cardiovascular health of an individual in response to different cellular stresses, including oxidative stress.

## 6. Regulation of Gene Expression by Small Non-Coding RNAs

In addition to directly regulating gene expression by acting on the components of the protein synthesis machinery, oxidative stress can also alter gene expression via small noncoding microRNAs (miRNAs). miRNAs, on average 22 nucleotides in length, are transcribed as precursors and processed into mature miRNAs capable of interacting with target cellular mRNA transcripts; in most cases, miRNAs induce target degradation (Figure 2D). miRNAs are critical for normal animal development by fine-tuning gene expression, while abnormalities in miRNA expression are associated with many human diseases (reviewed in [168]).

Numerous miRNAs are abundantly present in heart tissues and play an essential role in cardiac development and function [169]. For example, miR-1 (consisting of miR-1-1 and miR-1-2 isomers in humans) plays a vital role in cardiac and skeletal muscle proliferation and differentiation [170], while miR-133a (consisting of identical isomers miR-133a-1 and miR-133a-2) is indispensable in cardiomyocyte development. miR-133a is downregulated in patients suffering from myocardial infarction [171], linking miRNAs to CVDs. Another miRNA abundantly expressed in cardiac muscle cells is miR-208a, and its expression is necessary for correct cardiac functioning in mice [172] and cardiac remodeling in diabetic patients [173]. miR-499 is only present in cardiac cells, and regulates the late differentiation stages of human cardiomyocyte progenitor cells [174]. The expression of miR-499 modulates cardiomyopathy [175] and inhibits the hypoxia/reoxygenation-induced cardiomyocyte injury [176].

Interestingly, recent works have shown that oxidative modifications of some miRNAs altered their targets with significant implications for cardiovascular functioning. For example, oxidative modification of miR-184 allows it to target anti-apoptotic factors Bcl-xL and Bcl-w, which are not affected by the unmodified miR-184. This altered targeting by oxidized miR-184 increased the susceptibility of the mammalian heart to ischemia/reperfusion (I/R) injury in rat heart cell line H9c2 and mouse models [177]. Elevated expression of some miRNAs, on the contrary, has a cardioprotective role. Overexpression of miR-506 and miR-124 mitigated the effects of H_2_O_2_-induced human cardiomyocyte injury [178]. Similarly, increased expression of miR-19b reduced H_2_O_2_-induced apoptosis by targeting the phosphatase and tensin homolog (PTEN) gene in H9c2 rat heart cell lines [179]. Thus, miRNAs represent disease biomarkers and an attractive object for therapeutic intervention against cardiovascular diseases [180]. Recent studies have attempted to validate the therapeutic efficacy of some of miRNA-targeting oligonucleotides, like anti-miR-208a. Thus, it was shown that silencing miR-208a expression in the hearts of Dahl salt-sensitive rats improves cardiac function [181]. The efficacy of anti-miR-208a therapy was found to be dependent on the stage of disease [182].

In addition to miRNAs, recent work also showed a unique mechanism by which tRNA molecules undergo site-specific cleavage in response to stress, generating tRNA-derived RNA fragments or tRNA-derived stress-induced RNA. These novel classes of short tRNAs can modulate different biological functions [183], including inhibition of global protein synthesis via phosphorylation of eIF2α (see Section 4.1) [184], or act as cell damage markers [185]. Thus, several steps of the protein synthesis pathway can be targeted either directly or indirectly by these small non-coding RNAs, providing an additional layer of complexity to translational regulation of gene expression during oxidative stress.

## 7. Antioxidant Therapies for Cardiovascular Diseases

Given that oxidative stress contributes to etiology of various CVDs, antioxidant therapies have long been considered a promising strategy to treat and prevent these disorders [186]. Antioxidant treatment strategies can be divided into three major categories, the first of which includes natural defense systems composed of vitamins: vitamin C (ascorbic acid), vitamin E (tocopherols), vitamin A (retinol, retinal, and retinoic acid), and vitamin D (secosteroids). Recent studies demonstrated a positive outcome of vitamin treatment of hypertension [187], cardiac hypertrophy [188], and post-operative complications after heart surgery [189]. In a large-scale populational study, antioxidant vitamins were administrated to 875 patients for 22 years; increased dietary intake of vitamins A, C, and E was associated with decreased incidents of adverse cardiovascular abnormalities [190]. The second group of antioxidants, which includes dietary supplements like coenzyme Q10, carotenoids, and polyphenols, showed significant activity in managing ischemic heart disease, stroke, arrhythmia, atherosclerosis, and hypertension [11]. Resveratrol, a polyphenol found in grapes/red wine, plays a protective role by promoting anti-inflammatory responses through scavenging TNF-α-triggered ROS [191]. Despite encouraging results, these oral therapies have their limitations, including poor bioavailability of the antioxidants due to low solubility, low oral absorption, and instability in the gastrointestinal tract. In addition, multiple studies summarized by Cobley and co-authors provide compelling evidence that antioxidant properties of vitamins C and E are minimal, especially in neutralizing exercise-induced ROS [192]. This could be due to general inability of vitamins to target mitochondria (the major origin of exercise-induced ROS), poor ability to recognize ROS, and/or attenuation of NRF2 activation (see Figure 2 in [192]). Nevertheless, several excellent reviews discuss the benefits and limitations of oral regimens in treating CVDs [12,32,192]. The third strategy of antioxidant therapies is the antioxidant gene therapy that is based on delivering ROS-scavenging enzymes to the site of injury. They include heme oxygenase-1, extracellular superoxide dismutase, glutathione peroxidase, and catalase (reviewed in [193]). The main limitation of this kind of CVD therapy is the restricted efficacy of the delivery vectors available today [194].

Of particular interest for this review, several reports provide circumstantial evidences suggesting an effect of antioxidant therapies on translational control. For example, a plant-based polyphenolic compound polyproanthocyanidin, known for its antioxidant properties, was shown to interact with eIF2 and inhibit eIF2α phosphorylation [195]. Resveratrol positively affects cardiovascular health and has been shown to act through the eIF4F complex, suggesting that eIF4F-mediated regulation of translation initiation for specific messages might be involved in its cardioprotective role [196]. Similarly, intracardial administration of vitamin D in chickens reduced binding of Met-tRNA_i_^Met^ to ribosomal particles and interfered with 43S/48S complex formation [197]. Although these studies provide *prima facie* evidence suggesting modulation of specific gene expression by antioxidants, detailed studies are needed to better understand the mechanisms involved in this type of regulation and their implications on the cardiovascular well-being of an individual.

## 8. Concluding Remarks

The purpose of this review was to summarize the current knowledge on the effects of ROS on the cardiovascular system on the level of protein translation. As described, different kinds of abnormalities in protein synthesis are associated with a plethora of CVDs, some of which are mediated by ROS. However, most of the studies on this topic lack molecular constituents and remain mostly observational. With the progressive growth of whole-genome technologies, such as RNAseq, ribosome profiling, proteomics, redoxomes, and others, we are optimistic that molecular mechanisms of cardiovascular pathologies will be unraveled soon.

In addition, our understanding of the concept of oxidative stress has also changed over the years. Defined in 1985 by Helmut Sies as a disturbance in the balance between ROS and antioxidants [30], the term “oxidative stress” has since widened its meaning [31]. As such, the discovery that ROS can function as signaling molecules by transmitting messages from the environment or cellular organelles provide the mechanism for induction of a particular gene expression program that can occur transcriptionally and/or translationally. For example, the Cys-Cys-switch found in the Yap1 transcription factor master regulator of *Saccharomyces cerevisiae* leads to a conformational change upon Cys oxidation of Yap1, promoting the factor’s translocation into the nucleus to activate expression of ~100 stress-responsive genes [198]. Low-dose oxidative stress catalyzes chemical Fe(II)-dependent hydrolysis of 25S rRNA within the regulatory region of the yeast ribosome (expansion segment 7, ES7L), generating translationally active ribosomal species that likely participate in stress adaptation [199,200]. These examples demonstrate that ROS are not always damaging, but rather, in biological systems like eukaryotic cells, can play dual roles, adaptive at low concentrations and destructive at high concentrations. It is very important to apply this new knowledge when we investigate cardiovascular pathologies mediated by oxidative or other types of stress that generates ROS. In particular, it will be interesting to examine whether and how low-dose ROS participate in the mechanisms of ORFs selection, IRES versus 5ʹ cap-dependent translation initiation, regulation of translation factors activities via protein modifications, fidelity of the decoding process, and protein quality control.

## Figures and Tables

**Figure 1 ijms-21-02661-f001:**
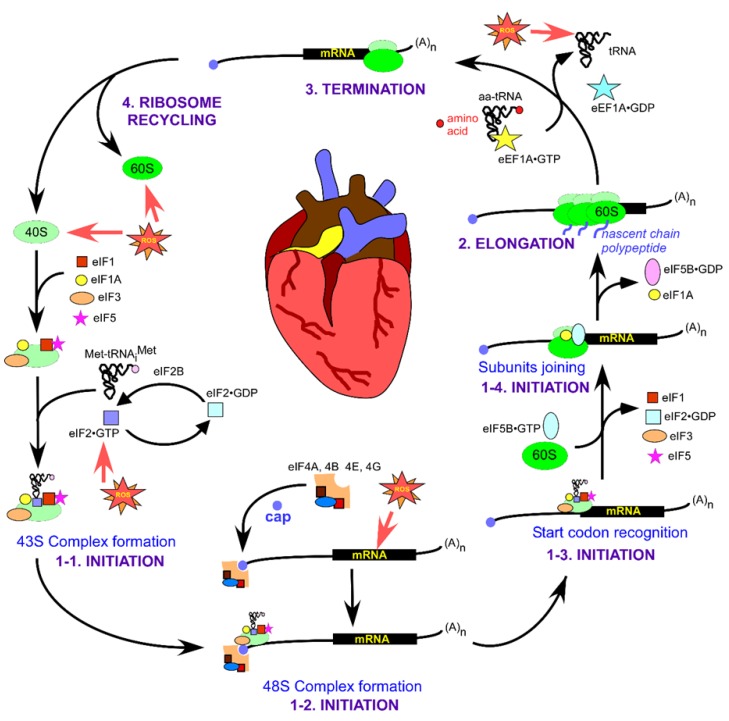
Protein translation in eukaryotes using the canonical pathway for translation initiation. The four stages of translation—INITIATION, ELONGATION, TERMINATION, and RECYCLING—are marked in purple. The canonical pathway of translation initiation is divided into four steps: 43S complex formation (1-1), 48S complex formation (1-2), start codon recognition (1-3), and joining of the ribosomal subunits (1-4). A detailed explanation of the process of eukaryotic translation is provided in Section 3. Reactive oxygen species (ROS) are shown as red stars. Known ROS targets within the cardiovascular system are marked with red arrows.

**Figure 2 ijms-21-02661-f002:**
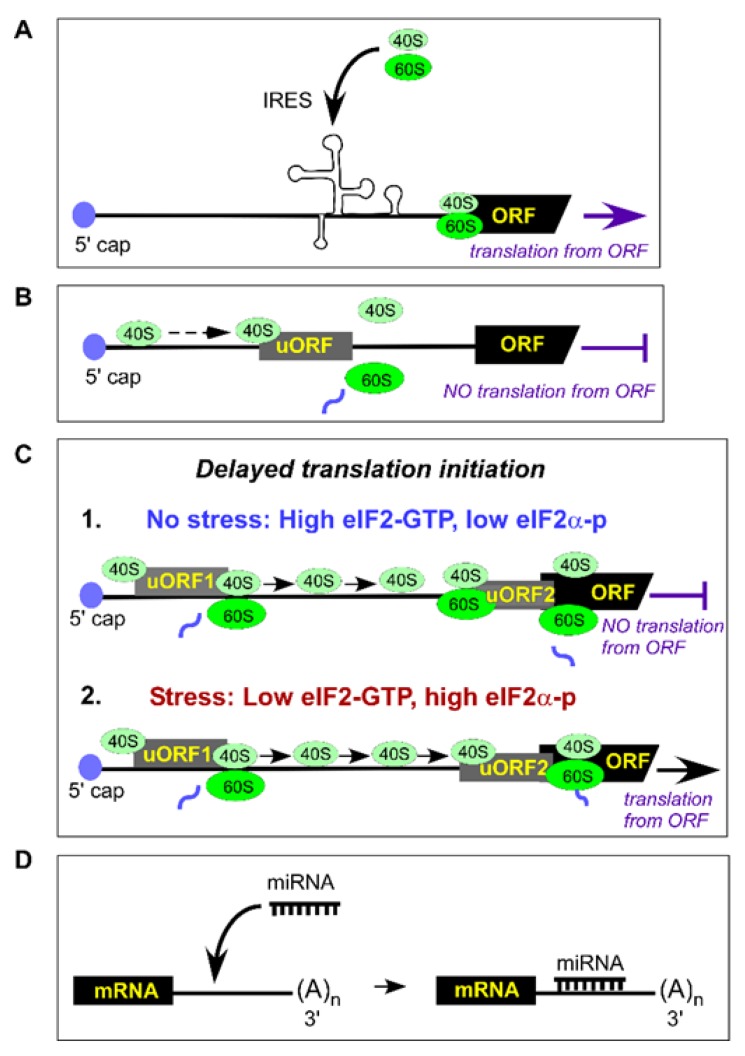
(**A**–**C**). Unconventional mechanisms of translation initiation in eukaryotes. (**A**) Internal ribosome entry site (IRES)-mediated translation initiation. (**B**,**C**) Upstream open reading frames (uORFs) regulate downstream mRNA translation. (**B**) Translation initiation from uORFs may downregulate the translation of the downstream sequence of the primary coding gene. The dotted arrow represents 40S ribosomal subunits scanning 5’UTR. (**C**) Delayed translation initiation. This stress-dependent mechanism of translation initiation is controlled by upstream uORFs and described in detail for *ATF4* in Section 4.2.2. The solid arrows represent re-initiating 40S subunits. The thick solid black arrows represent ongoing translation process, the thick solid purple arrows represent translational shut-down. (**D**) Translation regulation by small noncoding RNAs (miRNA), detailed in Section 6.

**Table 1 ijms-21-02661-t001:** Major types of radical and non-radical species.

**Radicals**	O_2_^•–^	Superoxide anion
	OH•	Hydroxyl
	NO•	Nitric oxide
	HOO•	Hydroperoxyl
**Non-radicals**	H_2_O_2_	Hydrogen peroxide
	HOCl	Hypochlorous acid
	ONOO^−^	Peroxynitrite
	^1^[O_2_]	Singlet oxygen

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
