# Peer review of "Effects of Oxidative Stress on Protein Translation: Implications for Cardiovascular Diseases"

_ijms, 2020, doi:10.3390/ijms21082661_

Round 1

Reviewer 1 Report

This work provides a high quality, and comprehensive review of oxidative stress in protein translation and implications for cardiovascular diseases. It highlights the novel concept that oxidative alterations of protein translation are implicated in cardiovascular diseases.

The following minor corrections are recommended.

Page 2: Add a sentence for the role of NADPH oxidase as a source of ROS and the known role of NADPH oxidase in CVD.

Page 2: Revise definition of oxidative stress to include multiple ROS sources rather than limit to mitochondrial ETC.

Page 8, 291: Revise “responded differently to different oxidative stressors” to something like “responded differently to distinct oxidative stressors”.

Author Response

This work provides a high quality, and comprehensive review of oxidative stress in protein translation and implications for cardiovascular diseases. It highlights the novel concept that oxidative alterations of protein translation are implicated in cardiovascular diseases. The following minor corrections are recommended.

We thank the Reviewer for the insightful evaluation of the manuscript and for finding our review article “high quality and comprehensive.” We are grateful for the suggestions for improving our work. Please find our point-by-point answers below. 

  • Page 2: Add a sentence for the role of NADPH oxidase as a source of ROS and the known role of NADPH oxidase in CVD.

Answer: Please see our answer to critique #2-2.     

  • Page 2: Revise definition of oxidative stress to include multiple ROS sources rather than limit to mitochondrial ETC.

Answer:

2-1. We agree with the Reviewers #1 that the definition of oxidative stress should be revised. Thus, a new paragraph has been incorporated into Section 2.3 of the revised version: “Nevertheless, in situations when defense systems are either overwhelmed or malfunction, upon abnormalities/mutations in ETC components, or during loss of mitochondrial inner membrane integrity, ROS levels exceed functional capacity of antioxidants (Strich, 2015), creating a condition called oxidative stress. In fact, the concept of oxidative stress was first formulated in 1985 as “a disturbance in the prooxidant-antioxidant balance in favor of the former” (Sies and Cadenas, 1985). Three decades of extensive research have built a body of evidence establishing oxidative stress as a complex and multifaceted physiological condition that can be divided into several sub forms, ranging from physiological to toxic and acute (for more details, please read an excellent review written by Helmut Sies and colleagues (Sies et al., 2017).”

2-2. Regarding multiple sources of ROS: we highlighted the difference between source and site of ROS production. The new paragraph reads: “ROS can be supplied from the environment or from within the cell. Exogenous sources include exposure to radiation or high levels of transition metals, heavy metals, and several anticancer drugs (Yokoyama et al., 2017). The main intracellular origin of ROS generation is the mitochondria, but other cellular sites, such as the endoplasmic reticulum (ER), peroxisomes, actin cytoskeleton, lysosome, plasma membrane, and cytosol are also known to supply ROS (reviewed in (Di Meo et al., 2016; Strich, 2015)). The main intracellular sources of ROS are the electron transport chain (ETC) of mitochondria (see below) and certain enzymes that generate ROS as reaction by-products. For example, xanthine oxidase (XO), which represents an oxidized form of xanthine dehydrogenase (XDH), forms in damaged tissues, including rat hearts with myocardial ischemia. XO can transfer electrons to molecular oxygen, resulting in the formation of superoxide during xanthine or hypoxanthine oxidation and promoting further heart damage (McCord et al., 1985). Another example of enzymes that supply ROS is NADPH oxidases (NOXs), a unique family of enzymes the only function of which is producing ROS. The five family members (NOX 1-5) share the same structural characteristics, including a common catalytic core composed of two domains: the dehydrogenase domain and the transmembrane domain that ensures membrane localization of these enzymes. Electrons are transferred from cytosolic NADPH to FAD located in the dehydrogenase domain, then to the inner and outer heme present in the transmembrane domain, and finally to oxygen located on the opposite site of the membrane, thus generating either superoxide (NOX1-3, NOX5) or H2O2 (NOX4) (reviewed in (Magnani and Mattevi, 2019)). NOXs are implicated in CVD pathologies, including myocardial ischemia-reperfusion injuries, myocardial damage, and cardiomyocyte death (Braunersreuther and Jaquet, 2012; Braunersreuther et al., 2013).” Changes are highlighted in red.      

  • Page 8, 291: Revise “responded differently to different oxidative stressors” to something like “responded differently to distinct oxidative stressors”.

Answer: Per your recommendation, we changed the sentence in line 418 (former 291) to: “Interestingly, TNN uORFs responded differently to distinct oxidative stressors.” The change is highlighted in red. 

Reviewer 2 Report

The authors provide a summary of the relationships between translational degradation, ROS and cardiovascular disease.
The subject of the connection between ROS and the pathophysiology of cardiovascular disease is an intensely published one and there are many review articles already published on this subject. But the authors do a good job by extending this topic and including translational protein degradation in the concept of cardiovascular disease pathophysiology.
The references indicate most recently published articles. Therefore the review is a useful entry point into current literature for those unfamiliar with this area.

Author Response

Thank you for the positive comments on our review article.

Reviewer 3 Report

The authors have produced a timely and interesting account of the role of oxidative stress in CVD with a specific focus on protein synthesis. They provide an excellent well-informed account of protein synthesis. I believe their work might be improved by considering some of the more nuanced aspects of redox biology. In particular, many of mechanisms linking specific species to defective protein synthesis via post-translational events (e.g., thiol modification) seem to be lacking. 

  • In the abstract, can reactive species really accumulate? Even if you added 1 uM of superoxide to just water, nevermind a biological system, it would disappear in seconds! Please consider Helmut Sies recent work when defining oxidative stress (e.g., https://www.annualreviews.org/doi/abs/10.1146/annurev-biochem-061516-045037)
  • A clear example of how “ROS” ideally with respect to a specific species can disrupt protein homeostasis with attendant implications for CVD should be provided in the abstract.
  • Please revise the definition of ROS….they are radical and non-radical derivatives of ground state molecular dioxygen, which is itself a di-radical, with heterogenous chemical reactivity in biological systems. Treating them as a single entity is perilous see https://www.ncbi.nlm.nih.gov/pmc/articles/PMC4445605/
  • Are the defence enzymes really thus when they can readily participate in redox signalling? indeed, even superoxide dismutase can be a pro-oxidant.
  • Univalent reduction of O2 to superoxide by cytochrome c oxidase is necessary to breath! COX merely used two proximal transition metals to trap free radicals. That is to say, it is possible for nature to evolve ETC components able to use ROS without realising them. Is it not therefore, possible that any “leak” from complex I and complex III…is intended? Could it not serve a signalling role that reports on mitochondrial function. See perhaps https://onlinelibrary.wiley.com/doi/abs/10.1002/bies.201100051
  • How do ROS impact eif2? At the molecular level?
  • Please elaborate the link between VEGF ROS and oxidative stress in hypoxia.
  • What ROS do ATF4 over expression induce? If the species is unknown, then this is a clear gap in understanding that could be critiqued.
  • RE titin consider the recent work of Chouchani on oxidative stress in the older heart as mediated by reversible thiol oxidation https://www.ncbi.nlm.nih.gov/pubmed/32109415
  • How do ROS impact TGFB activity? The molecular details are either lacking or undescribed.
  • Until the exact nature of the oxidative stress is disclosed, it will be difficult to rationally treat CVD. The inability of the antioxidant listed to target mitochondria should be considered, as well as their chemical strictures. Many fail to react with the key species see https://www.sciencedirect.com/science/article/pii/S2213231718300041 and https://www.sciencedirect.com/science/article/pii/S0891584915001422
  • The study of Topf and colleagues may be useful https://www.nature.com/articles/s41467-017-02694-8
